# Hydrocracking of Octacosane and Cobalt Fischer–Tropsch Wax over Nonsulfided NiMo and Pt-Based Catalysts

**Wenping Ma [1,\*], Jungshik Kang [1,2], Gary Jacobs [1,3,4], Shelley D. Hopps [1] and Burtron H. Davis [1]**

[1] Center for Applied Energy Research, University of Kentucky, 2540 Research Park Drive, Lexington, KY 40511, USA; jungshik.kang@udri.udayton.edu (J.K.); gary.jacobs@utsa.edu (G.J.); shelley.hopps@uky.edu (S.D.H.); burtron.davis@uky.edu (B.H.D.)

[2] University of Dayton Research Institute, 300 College Park, Dayton, OH 45469, USA

[3] Department of Biomedical Engineering and Chemical Engineering, University of Texas at San Antonio, One UTSA Circle, San Antonio, TX 78249, USA

[4] Department of Mechanical Engineering, University of Texas at San Antonio, One UTSA Circle, San Antonio, TX 78249, USA

\* Correspondence: wenping.ma@uky.edu; Tel.: +1-859-257-0331

**Abstract:** The effect of activation environment ($N_2$, $H_2$ and $H_2S/H_2$) on the hydrocracking performance of a NiMo/Al catalyst was studied at 380 °C and 3.5 MPa using octacosane ($C_{28}$). The catalyst physical structure and acidity were characterized by BET, XRD, SEM-EDX and FTIR techniques. The $N_2$ activation generated more active nonsulfided NiMo/Al catalyst relative to the $H_2$ or $H_2S$ activation ($X_{C28}$, 70–80% versus 6–10%). For a comparison, a NiMo/Si-Al catalyst was also tested after normal $H_2$ activation and showed higher activity at the same process conditions ($X_{C28}$, 81–99%). The high activity of the NiMo/Al ($N_2$ activation) and NiMo/Si-Al catalysts was mainly ascribed to a higher number of Brønsted acid sites (BAS) on the catalysts. The hydrocracking of cobalt wax using Pt/Si-Al and Pt/Al catalysts confirmed the superior activity of the Si-Al support. A double-peak product distribution occurred at $C_4$–$C_6$ and $C_{10}$–$C_{16}$ on all catalysts, which illustrates secondary hydrocracking and faster hydrocracking at the middle of the chain. The nonsulfided NiMo/Al and Pt/Al catalysts, and NiMo/Si-Al catalyst produced predominantly diesel (sel. 50–70%) and gasoline range (sel. > 50%) hydrocarbons, respectively, accompanied by some $CH_4$ and light hydrocarbons $C_2$–$C_4$. On the other hand, the hydrocarbon distribution of the Pt/Si-Al varied with conditions (i.e., diesel sel. 87–90% below 290 °C or gasoline sel. 60–70% above 290 °C accompanied by little $CH_4$) The dependence of the isomer/paraffin ratio on chain length was studied as well. The peak iso/paraffin value was observed at $C_{10}$–$C_{13}$ for the SiAl catalyst.

**Keywords:** hydrocracking; isomerization; $N_2$ activation; NiMo; Pt; $Al_2O_3$ (Al); $SiO_2$-$Al_2O_3$ (Si-Al); octacosane; Fischer–Tropsch wax

## 1. Introduction

Hydrocracking is a key process in petroleum refining for the conversion of heavy hydrocarbons into a variety of high-value fuels such as gasoline ($C_5$–$C_{11}$), jet fuel ($C_8$–$C_{16}$), and middle distillates ($C_{10}$–$C_{20}$). In particular, the hydrocracking process is of great importance to meet the growing demand for high quality diesel fuels with stringent environment specifications [1,2]. Generally, hydrocracking is conducted over metal/acid bifunctional catalysts [1,3–9]. The most conventional hydrocracking catalysts are noble metal (Pt and Pd) catalysts [10] and non-noble metal catalysts such as NiMo, NiCo, CoMo, NiW, WZr catalysts [11–16]. The balance between metal sites and acid sites in supports is crucial for the catalyst performance, as are the support physical properties (e.g., pore size and BET surface area) [13,16–20]. As such, the effect of support type has been extensively investigated, and examples include amorphous oxides or mixtures of oxides (e.g., $Al_2O_3$, $SiO_2$–$Al_2O_3$, $ZrO_2/SO_4^{2-}$), Y-, beta zeolites, ZSM-5, ZSM-22, Al-MCM-41 zeolite and

SAPO 11, SAPO 31, SAPO 34 and SAPO 41 silicoaluminaphosphates [3]. Considerably higher diesel selectivity and higher isomerization selectivity have been obtained using supported noble metal (e.g., Pt and Pd) catalysts with supports containing moderate Brønsted acidity and large pore size [10,13]. The strength of the Brønsted acid sites was reported to follow the order: $SiO_2 < \gamma\text{-}Al_2O_3 < SiO_2\text{-}Al_2O_3 < H\text{-}ZSM\text{-}5$ [21].

The non-noble metal hydrotreating catalysts such as NiMo, NiCo, CoMo, and NiW have received extensive attention in past decades due to their lower costs relative to the noble metal catalysts [11–16]. These catalysts were primarily used in sulfided form in the refining process of petroleum [1,2,11,13,22], in which up to several hundreds of ppm S could be present. From an environmental point of view, nonsulfided hydrocracking catalysts are more preferable; furthermore, the nonsulfided catalysts are suitable for upgrading commercial Fischer–Tropsch synthesis (FTS) wax that is free from, or contains extremely low amounts of sulfur and aromatics. However, much less emphasis has been placed on the nonsulfided hydrotreating catalysts, especially for nickel-containing catalysts [13]. Haan et al. [13] studied the hydrocracking of iron Fischer–Tropsch (FT) wax and $C_{16}$ over nonsulfided 3.0–4.5%Ni/Si-Al (Sirolax 40) catalyst by employing $H_2$ activation. A diesel fraction of 75% was obtained at 360 °C and 7.0 MPa, and larger Ni particles on 4.5%Ni catalyst yielded higher selectivity to $C_1$–$C_2$ light hydrocarbons. To date, little study was done with nonsulfided NiMo on alumina catalyst. Therefore, more efforts are needed to study the nonsulfided Ni-Mo/Al catalyst for better understanding the catalyst performance and reaction mechanism. Herein, we further investigated a nonsulfided commercial NiMo/Al (AZKO) catalyst by using $N_2$ activation in addition to the traditional S and $H_2$ activation strategy.

In addition to less fundamental studies on nonsulfided Ni hydrocracking catalysts, some early results about the isomer content is subject to debate. Leckel et al. [23] studied diesel selective hydrocracking of $C_{15}$–$C_{45}$ FTS wax over a 1.2%Pt/Si-Al catalyst using a bench-scale trickle bed reactor at 350–380 °C. The isomerization selectivity was found to be inhibited at a high pressure of 7.0 MPa. Meanwhile, the isomerization selectivity increased with carbon number, peaking at $C_{13}$; thereafter, the selectivity for isomer formation decreased noticeably. Beyond carbon number 22, the iso-paraffin-to-n-paraffin ratio continuously decreased with increasing chain length but with a much flatter slope (Figure 1a). In the study on hydrocracking of FT wax by Gamba et al. under similar reaction conditions, i.e., 340–370 °C, 3.5–6.0 MPa and $H_2$/wax = 12–30; GHSV of 1–3 $h^{-1}$ over a Pt/Si-Al catalyst [10], the iso-normal ratio versus chain length in the products showed a different pattern. The iso/total ratio first decreases with carbon number until $C_{10}$; afterwards, it increases with increasing chain length up to $C_{22}$ (Figure 1b). Therefore, additional study is necessary to elucidate the issue of the dependence of isomer selectivity on chain length, which can significantly affect the fuel quality (e.g., cloud point and cetane number).

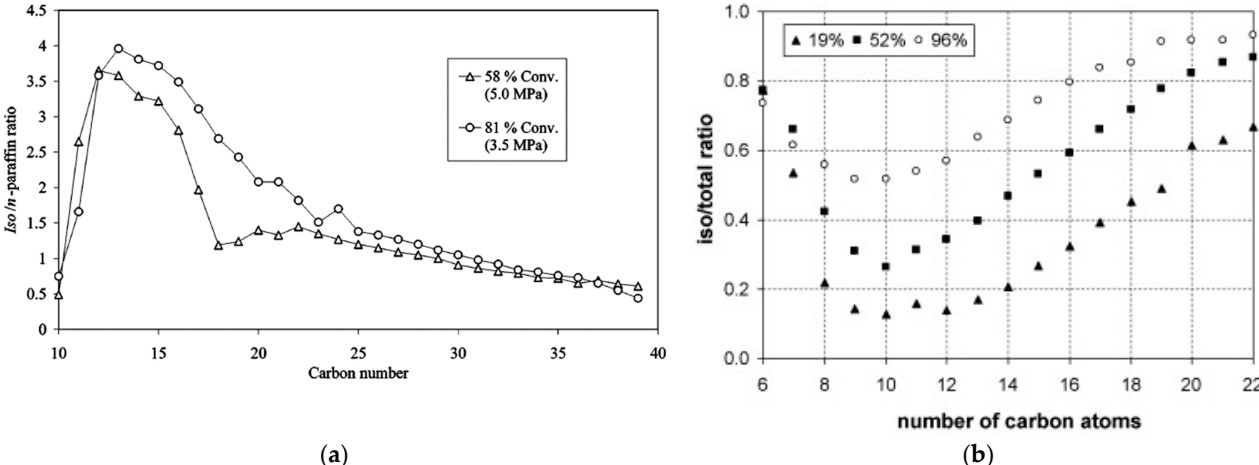

(a)  (b)

**Figure 1.** Isoalkane/alkanes ratio versus chain length: (**a**) reproduced with permission from [23] Copyright 2006, ACS; (**b**) reproduced with permission from [10] Copyright 2010, Elsevier.

For the hydroconversion of hydrocarbons, the reaction mechanism occurring on metal/acid catalysts is the subject of debate, even though numerous related studies have been performed. It is generally accepted that hydroconversion of long chain hydrocarbons could undergo three competitive processes: hydroisomerization, catalytic hydrocracking, and thermal cracking. Thermal cracking under hydrogen atmosphere, in which the carbon-carbon bond breakage was assumed to proceed via a free radical mechanism, was proposed by Rice et al. [24,25]. The carbonium mechanism that was earlier used to explain the hydrocarbon distribution over Si-Al catalysts by Thomas [26] and Greensfelder et al. [27], has been widely accepted in the field [5–9]. For example, on a bifunctional catalyst, it is completed via alkane dehydrogenation, carbonium ion formation, alkyl shift, and hydrogenation. Despite the fact that the classical carbonium ion mechanism on acid sites with Si-Al support [11,27–30] has been studied extensively, the question of whether metals such as Ni, Mo, Pt, and $Al_2O_3$ support can influence the reaction pathways has not been well addressed. This study provides insight into the reaction pathways occurring during hydrocracking over different metal catalysts.

As such, we have three objectives in this study by choosing $Al_2O_3$-supported NiMo catalyst: (1) to investigate the hydrocracking performance of a nonsulfided commercial NiMo/Al made by $N_2$ activation. For the purpose of comparison, the NiMo/Al catalyst showing poor activity under $H_2$ and $H_2S$ activation, a different NiMo/Si-Al catalyst showing very high activity under normal $H_2$ activation are also investigated; (2) to establish the relationship between iso-paraffin content versus chain length on the $Al_2O_3$ and Si-Al supported NiMo and Pt catalysts; and (3) to shed light on the reaction mechanism occurring on various Pt and NiMo catalysts.

## 2. Materials and Methods

### 2.1. Material and Catalyst Preparation

The $NiMo/Al_2O_3$ catalyst (2.5%Ni12.9%Mo, NiMo/Al) used for the activation study was supplied by Akzo Nobel company (Akzo Nobel Naamloze Vennootschap (N.V.), Amsterdam, The Netherlands), while all other NiMo and Pt catalysts prepared at The University of Kentucky Center for Applied Energy Research (UK-CAER). The $\gamma$-$Al_2O_3$ support used in this work was Sasol Catalox-150 $\gamma$-$Al_2O_3$ (Sasol, Sandton, South Africa). The support was first calcined at 400 °C in a muffle furnace for 4 h before impregnation of metals. The silica–alumina support (Davison 135, Si/Al = 6.7) was purchased from William Russell (W.R. Grace and Company, Columbia, MD, USA). The $3\%Ni12\%Mo/Al_2O_3$ and $3\%Ni12\%Mo/SiO_2$-$Al_2O_3$ (NiMo/Si-Al) catalysts were prepared using incipient wetness impregnation (IWI). The Ni and Mo precursors are nickel (II) nitrate hexahydrate and ammonium molybdate tetrahydrate (Fisher Scientific International, Inc., Pittsburgh, PA, USA), they were sequentially impregnated. The $0.5\%Pt/Al_2O_3$ (Pt/Al) and $0.5\%Pt/SiO_2$–$Al_2O_3$ (Pt/Si-Al) catalysts were prepared by IWI as well. Tetraammine platinum (II) nitrate (Pt 50.38%) (Alfa Aesar, Haverhill, MA, USA) was used as the platinum precursor. Following each impregnation, the catalysts were dried at 110 °C for 12 h, and then they were further calcined in air at 500 °C for 4 h.

### 2.2. BET Measurement

The surface area, pore volume, and average pore radius of the supports and catalysts were measured by BET using a Micromeritics Tri-Star 3000 (Micromeritics, Norcross, GA, USA) gas adsorption analyzer. Approximately 0.2 g of sample were weighed and loaded into a 3/8″ quartz tube. Nitrogen (Scott-Gross Company, Lexington, KY, USA) was used as the adsorption gas and sample analysis was performed at the boiling temperature of liquid nitrogen. Prior to the measurement, the sample was slowly ramped to 433 K and evacuated overnight to approximately 6.7 Pa.

### 2.3. X-ray Diffraction (XRD)

X-ray diffraction (XRD) on powder NiMo and Pt samples was performed at room temperature using a Rigaku DMAX-B Diffractometer (Panalytical X-Pert Pro, Malvern, UK) operating with Cu Kα radiation (1.54 Å), in order to identify catalyst structure. Prior to the XRD measurement, the NiMo/Al sample was reduced under different gas environments ($H_2$, $N_2$ and $H_2S$), while others (NiMo/Si-Al, Pt/Si-Al, and Pt/Al) were reduced at $H_2$, at 400 °C for 14 h, followed by passivation using a $1\%O_2/N_2$ gas mixture (Scott-Gross Company, Lexington, KY, USA).

### 2.4. Scanning Electron Microscopy (SEM-EDX)

To examine the morphology and composition of the catalysts, scanning electron microscopy with energy dispersive X-ray (SEM-EDX) studies were conducted on the Pt/Al, Pt/Si-Al and NiMo/Si-Al catalysts using a Hitachi S-4800 microscope (Hitachi, Limited (Ltd.), Tokyo, Japan) equipped with an X-ray energy-dispersive (EDX) microanalyzer (Oxford Instruments, Abingdon, Oxford Shire, UK). Ni, Mo, Pt, Al, Si, and O in the catalysts were detected by the EDX detector. The working parameters for the SEM and EDX used were 10 kV, 10 μA, and working distances (WD) of 10 and 15 mm, respectively. Prior to the experiment, the powder catalyst samples were mounted on sample stubs using carbon adhesive tape, and then sputter coated with gold for 90 s in a plasma coater (Hummer, Anatech Ltd., Battle Creek, MI, USA) to reduce charging effects.

### 2.5. FTIR Pyridine Adsorption

A thermo Nicolet Nexus 870 FT-IR spectrometer (ThermoFisher Scientific, Waltham, MA, USA) was employed to study Lewis and Brønsted acidity of the NiMo and Pt catalysts. A diffuse reflectance IR cell (Spectra-Tech, ThermoFisher Scientific, Waltham, MA, USA) equipped with a sample chamber that allow to flow gas and adjust temperature, was used for the diffuse reflectance infrared Fourier transform (DRIFT) experiment. The instrument was operated in the absorption mode at a resolution of $4\ cm^{-1}$. The catalysts were activated at 400 °C under $H_2$ or $N_2$ (Scott-Gross Company, Lexington, KY, USA) for 14 h in the same fixed bed reactor used for hydrocracking reaction. Then, the activated catalysts were passivated with 1% oxygen gas balanced with He (Scott-Gross Company, Lexington, KY, USA). Before taking background spectrum, the passivated catalysts were reactivated in the DRIFT cell at 400 °C under the same $H_2$ or $N_2$ condition for 1 h and cooled to a desired temperature 200 °C for adsorption. The pyridine adsorption under helium carrier gas was performed in the IR cell, and allowed equilibrate for 5 min.

### 2.6. Hydrocracking of $C_{28}$ and Cobalt FT Wax in a Fixed-Bed Reactor

Hydrocracking of n-octacosane (n-$C_{28}$) or FT wax produced using a conventional cobalt-based catalyst was performed in a homemade fixed-bed (FB) reactor made from 314 stainless steel (SS). The reactor is of 1 inch inner diameter and 22 inches in length. Approximately 1 g catalyst was diluted with 5 g glass beads with the size of 2 mm. Ultra high purity $H_2$ (Scott-Gross Company, Lexington, KY, USA) gas from a cylinder was passed through a reducing valve body and the flow was adjusted by a pre-calibrated Brooks (Brooks Instrument, Hatfield, PA, USA) mass flow controller before entering the reactor. $C_{28}$ or FT wax produced using a cobalt-based catalyst was melted in a 600 mL heating vessel first and suctioned into a high temperature ISCO syringe pump (Teledyne ISCO, Teledyne Technologies, Thousand Oaks, CA, USA). The hot liquid wax (130 °C) was pumped to the FB reactor at a desired flow rate through a tape-heated line using a syringe pump. To assess the performance of the nonsulfided NiMo/Al catalyst, the catalyst was activated in $N_2$ (Scott-Gross Company, Lexington, KY, USA) environment at 400 °C, 12 Nl/h, and 1 atm for 14 h. The activation of the NiMo/Al catalyst using $H_2$ or $H_2S$ (Scott-Gross Company, Lexington, KY, USA) at the otherwise same conditions was also performed for the purpose of comparison. All other NiMo/Si-Al, Pt/Si-Al, and Pt/Al catalysts were activated using $H_2$ before starting the hydrocracking reaction. After activation, the reactor temperature

was slowly dropped to the reaction temperature (i.e., within the range of 280–380 °C) and the reactor pressure was increased to 450 psig. During the hydrocracking time period, the flowrate of $H_2$ was adjusted to give a range of $H_2/C_{28}$ or $H_2$/wax molar ratios between 5 and 40. After exiting the reactor, the vapor passed through warm (170 °C) and cold (0 °C) traps to condense the liquid products. Gas flows were measured using a digital flow meter; reaction tailgases were collected in a 1-L gas sample bag and were analyzed using a Micro3000 GC (Agilent Technologies, Santa Clara, CA, USA). Liquid samples were collected and analyzed using a HP 5890 GC and a HP 5790 GC (Hewlett-Packard Company, Palo Alto, CA, USA). Generally, the sampling interval between the two samplings was ca. 24 h. The mass balance for the cracking runs fall within the range of 95–102%. In this study, $C_{28}$ conversion, wax $C_{20+}$ conversion and selectivity to various ranges of hydrocarbons are defined as follows,

$\quad\quad$ $C_{28}$ conversion (%):

$C_{28}$ conversion = $100 \times (C_{28}$ in feed $- C_{28}$ in product$)/C_{28}$ in feed

$\quad\quad$ Hydrocarbon selectivity (%):

$C_1 = 100 \times C_1$ in product/$(C_{28}$ in feed $- C_{28}$ in product$)$
$C_2–C_4 = 100 \times C_2–C_4$ in product/$(C_{28}$ in feed $- C_{28}$ in product$)$
$C_5–C_{11} = 100 \times C_5–C_{11}$ in product/$(C_{28}$ in feed $- C_{28}$ in product$)$
$C_{12}–C_{19} = 100 \times C_{12}$ in product/$(C_{28}$ in feed $- C_{28}$ in product$)$
$C_{20}–C_{27} = 100 \times (C_{20}–C_{27}$ in product $- C_{20}–C_{27}$ in feed$)/(C_{28}$ in feed $- C_{28}$ in product$)$

$\quad\quad$ Wax ($C_{20+}$) conversion (%) and selectivity to various ranges of hydrocarbons (%) are defined analogously

$C_{20+}$ conversion = $100 \times (C_{20+}$ in feed $- C_{20+}$ in product$)/C_{20+}$ in feed

$\quad\quad$ Hydrocarbon selectivity:

$C_1 = 100 \times C_1$ in product/$(C_{20}$ in feed $- C_{20}$ in product$)$
$C_2–C_4 = 100 \times C_2–C_4$ in product/$(C_{20}$ in feed $- C_{20}$ in product$)$
$C_5–C_{11} = 100 \times (C_5–C_{11}$ in product $- C_5–C_{11}$ in feed$)/(C_{20}$ in feed $- C_{20}$ in product$)$
$C_{12}–C_{19} = 100 \times (C_{12}–C_{19}$ in product $- C_{12}–C_{19}$ in feed$)/ (C_{20}$ in feed $- C_{20}$ in product$)$

## 3. Results

### 3.1. Characterization Results

3.1.1. BET Measurement Results

Table 1 summarizes the results of BET measurements for the $\gamma$-$Al_2O_3$ (Al) and $SiO_2$-$Al_2O_3$ (Si-Al) support, and the NiMo/Si-Al, NiMo/Al, Pt/Si-Al, and Pt/Al catalysts. The Si-Al support and the supported catalysts show much higher BET surface areas (395 $m^2$/g) than the $Al_2O_3$ support and catalysts (150 $m^2$/g). Addition of 15% NiMo on the Si-Al support results in decreasing the support surface area (289 $m^2$/g), similarly, addition of 0.5% Pt on the Al and Si-Al support lowered surface areas of the supports. The pore size of these catalysts displayed a similar trend. The NiMo/Al (AKZO) catalyst features with high surface area, 192 $m^2$/g, but similar pore volume to the Pt/Al catalyst. The actual surface areas of the NiMo and Pt catalysts are lower than the estimated values (310–390 $m^2$/g or 148 $m^2$/g) assuming that the support was the only contributor to the surface area. The BET results thus indicate that some pores of the Al and Si-Al supports were blocked by Pt or NiMo particles.

**Table 1.** Physical properties of supports, NiMo and Pt catalysts.

| Description | N₂ Physisorption | | |
|---|---|---|---|
| | BET Surf. Area m²/g | Pore Volume cm³/g | Average Pore Diameter, nm |
| Catalox-150 γ-Al₂O₃ (Al) | 150 | 0.5 | 10.5 |
| SiO₂-Al₂O₃ (Si-Al) | 395 | 0.66 | 7.0 |
| 0.5%Pt/Al | 145 | 0.42 | 9.6 |
| 0.5%Pt/Si-Al | 358 | 0.60 | 6.5 |
| 2.5%Ni12.9%Mo/Al [a] | 192 | 0.40 | 7.6 |
| 3%Ni12%Mo/Si-Al | 289 | 0.52 | 6.8 |

[a] reference [31].

### 3.1.2. X-ray Diffraction (XRD)

To investigate the impact of the activation environments (i.e., $H_2$, $N_2$ or $H_2S/H_2$), on the NiMo/Al catalyst structure, XRD characterization of an analogous NiMo/Al (3%Ni12%Mo) catalyst with similar Ni and Mo compositions to the AKZO catalyst was performed. For the purpose of comparison, the NiMo/Si-Al, Pt/Si-Al and Pt/Al catalysts after $H_2$ activation were also characterized. The XRD results for NiMo and Pt catalysts are shown in Figure 2A,B, respectively. In both plots, the diffraction peaks occurred at $2\theta$ of 32.8°, 37.1°, 39.5°, 45.6°, 60.7°, and 67° and can be indexed with the $\gamma$-$Al_2O_3$ phase having different planes such as (220), (311), (222), (400), (511), and (440) [32,33]. All XRD patterns of the different catalysts do not present notable sharp characteristic peaks for the Ni, Mo and Pt oxides or metal phases, which suggests either highly dispersed Ni, Mo or Pt dispersed on the supports or that the metal loadings were low. However, some small broad peaks representing fine metals or metal oxides are observed in the patterns; for example the peaks at 23–29° are attributed to $MoO_2$ and $MoO_3$ [34,35] for the $N_2$-activated NiMo/Al catalyst. The peaks at 13.7° and 59.8° are assigned to the $MoS_2$ phase [36] for the $H_2S/H_2$-treated NiMo/Al catalyst. Additionally, peaks at 44.5° and 50.5° are ascribed to Ni [37] for the NiMo/ or Ni/Si-Al catalysts, whereas the peak at 40° is assigned to Pt [38] for the Pt/Si-Al catalyst. These XRD results suggest the presence of fine $MoO_x$ (x = 2–3), $MoS_2$ and $Mo°$, $Ni°$, and $Pt°$ metals in the $N_2$-activated or $H_2S/H_2$-activated NiMo/Al catalysts and in the NiMo, Ni, or Pt on Si-Al catalysts, respectively. Furthermore, the XRD patterns of Si-Al supported NiMo or Pt catalysts shown in Figure 2A,B present a characteristic peak of $SiO_2$ at 23° [39], but no $Al_2O_3$ peaks show up, suggesting that $Al^{3+}$ ions are highly dispersed in the $SiO_2$ framework.

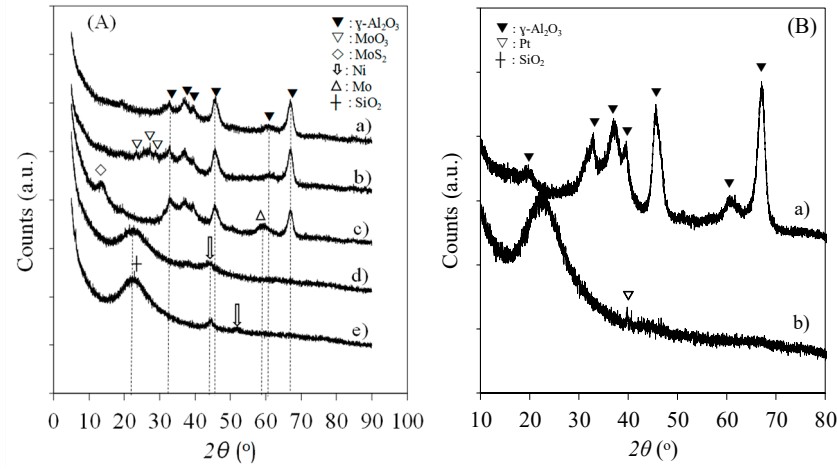

**Figure 2.** (**A**) XRD pattern of (a) $H_2$ reduced NiMo/Al catalyst, (b) $N_2$ reduced NiMo/Al catalyst, (c) 10%$H_2S/H_2$ reduced NiMo/Al catalyst, (d) $H_2$ reduced Ni-Mo/Si-Al catalyst, (e) $H_2$ reduced Ni/Si-Al catalyst; (**B**) XRD pattern of (a) $H_2$ reduced Pt/Al catalyst and (b) $H_2$ reduced Pt/Si-Al catalyst.

In summary, the XRD results suggest highly dispersed Ni, Mo or Pt metal or oxides on the $Al_2O_3$ or $SiO_2$-$Al_2O_3$ support. $N_2$ activation resulted in partially oxidized Mo ($MoO_x$, x = 2–3) in the NiMo/Al catalyst, which should be responsible for higher surface densities of Brønsted acid sites (BAS) on the catalyst (to be discussed in Section 3.1.4), and consequently higher hydrocracking activity (to be discussed in Section 3.2). Therefore, $N_2$ activation is an effective approach to create active nonsulfided NiMo/Al catalyst for hydrocracking hydrocarbons. The use of 10%$H_2S$/$H_2$ and $H_2$ treatments did not result in a significant difference in the catalyst structure, save for tiny $MoS_2$ and Mo peaks being observed for the sulfided NiMo catalyst.

### 3.1.3. Scanning Electron Microscopy (SEM-EDX)

Figures 3 and 4 illustrate the SEM micrographs and EDX results of the two Al and Si-Al supported Pt catalysts, respectively. The SEM images obtained at low ($\times 400$–800) and high ($\times 5$–20 K) magnification for each sample are provided in order to have a clearer picture of the support morphology. The particles for each support are irregularly shaped and randomly distributed. The surface of both Al and Si-Al supports based on the high magnification images of the catalysts are quite rough, some smaller particles are attached to larger particles, and multiple layers/steps, edges of the supports, and even some macro- and mesopores are observed in both images. Furthermore, fine metal particles reflected by tiny white spots can be observed everywhere, indicating highly dispersed metals on the surfaces of both supports. These are essential structures for generating high activity. The EDX detected all desired elements (Al, Si, O, Pt) on the supports (Figures 3c and 4c). Similar SEM images of the NiMo/Si-Al catalyst was also obtained, but are not shown for the sake of brevity.

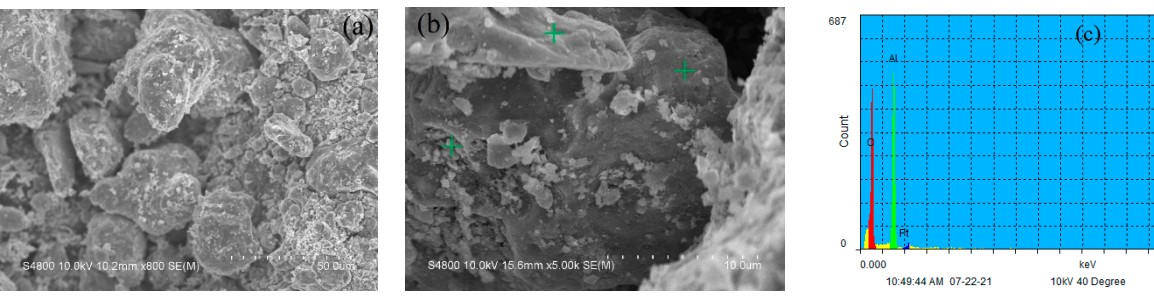

**Figure 3.** SEM image of Pt/Al catalyst (**a**) low magnification $\times 800$, (**b**) high magnification $\times 5$ k, (**c**) EDX spectrum.

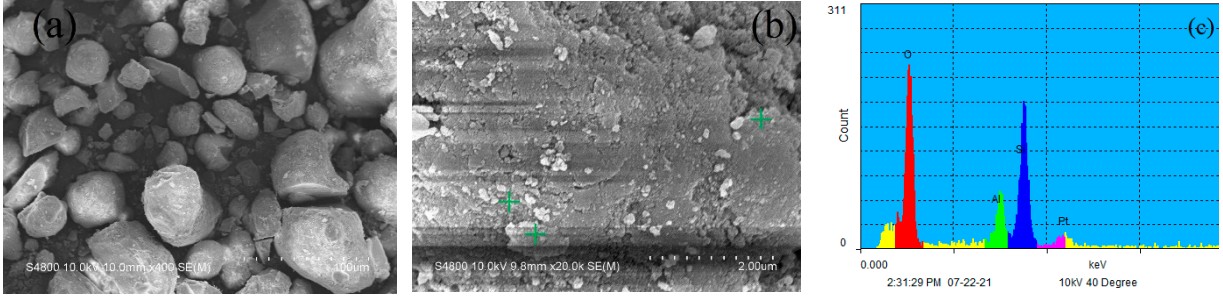

**Figure 4.** SEM image of Pt/Si-Al catalyst (**a**) low magnification $\times 800$, (**b**) high magnification $\times 5$ k, (**c**) EDX spectrum.

### 3.1.4. FTIR Pyridine Adsorption

Figure 5 presents the FTIR spectra of pyridine adsorption at 200 °C in the 1300–1700 $cm^{-1}$ region for the NiMo/$Al_2O_3$, Pt/$Al_2O_3$ and Pt/Si-Al catalysts activated in $H_2$ environment. The FTIR spectrum of the nonsulfided NiMo/$Al_2O_3$ catalyst activated by $N_2$ is also included. Consistently, all FTIR spectra show predominant bands at 1620 and 1450 $cm^{-1}$, which can be attributed to Lewis-bonded pyridine or Lewis acid sites (LAS). The presence of relatively

small bands at 1540 and 1640 cm$^{-1}$ indicates the presence of Brønsted acid sites (BAS). This is in line with the B and L acid sites identified in the studies of [17,40]. More interestingly, the spectra of the N$_2$-activated NiMo/Al catalyst and the Pt/Si-Al catalyst show higher intensity 1540 and 1640 cm$^{-1}$ bands relative to the other samples, suggesting much higher relative amounts of BAS in the samples. The higher BAS density of the Pt/Si-Al catalyst is due to the nature of the Si-Al acidic support; however, the higher amount of BAS for the N$_2$-activated NiMo/Al catalyst must be due to the N$_2$-activation procedure. We did not run the FTIR pyridine adsorption on the NiMo/Si-Al and the H$_2$S-treated NiMo/Al$_2$O$_3$ catalysts, but the acidity results on the two samples should be similar to that of the Pt/Si-Al (high BAS), and the H$_2$ activated NiMo/Al (low BAS), respectively. Puello-Polo et al. [40] performed a pyridine adsorption study over a sulfided NiMo/Al catalyst treated by H$_2$S, and the NiMo/Al essentially produced similar FTIR spectra pattern with strong L site peaks at 1460 and 1620 cm$^{-1}$ and tiny B peaks at 1540 and 1640 cm$^{-1}$ [40]. The FTIR pyridine characterization results indicate that the nonsulfided Al$_2$O$_3$ and Si-Al supported NiMo and Pt catalysts possess both LAS and BAS, but the amount of BAS in terms of the estimated B/L ratio follows the trend: Si-Al catalyst > N$_2$ activated NiMo/Al > H$_2$ activation > H$_2$S activation NiMo/Al. Therefore, N$_2$-activation created higher amounts of BAS on the NiMo/Al$_2$O$_3$ catalyst, which should be a major reason for the higher hydrocracking activity (to be discussed). The FTIR results appear to be consistent with the XRD results, which indicates the formation of some fine partially oxidized molybdenum (MoO$_x$) that may serve as a type of BAS.

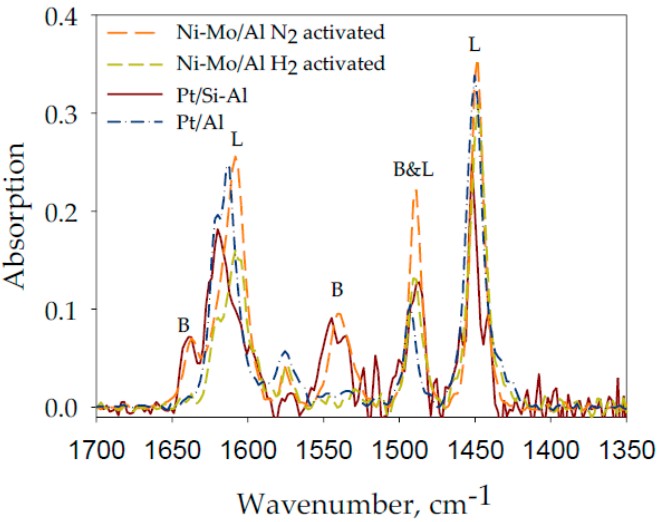

**Figure 5.** FTIR spectra of pyridine adsorbed over nonsulfided NiMo/Al$_2$O$_3$, Pt/Al$_2$O$_3$ and Pt/Si-Al catalyst at 200 °C.

### 3.2. Hydrocracking C$_{28}$ over Nonsulfided NiMo/Al and NiMo/Si-Al Catalysts

The hydrocracking performance of the nonsulfided NiMo/Al catalyst made by N$_2$-activation, was evaluated using C$_{28}$. Meanwhile, the non-sulfided NiMo/Al catalyst made by H$_2$-activation, the sulfided Ni-Mo/Al catalyst obtained by using 10%H$_2$S/H$_2$ pretreatment, and the nonsulfided NiMo/Si-Al catalyst pretreated by H$_2$, were also employed for hydrocracking. The hydrocracking results under the conditions 280–380 °C, H$_2$/wax = 20, LHSV = 1–2 h$^{-1}$ and 3.16 MPa are shown in Table 2 and Figures 6 and 7.

**Table 2.** Summary of hydrocracking $C_{28}$ on the NiMo/Al (AZKO) and NiMo/Si-Al catalysts [a].

| Catalyst | NiMo/Al | | | | | | NiMo/Si-Al | | |
|---|---|---|---|---|---|---|---|---|---|
| Activation | $N_2$ | $N_2$ | $N_2$ | $H_2S$ | $H_2S$ | $H_2$ | $H_2$ | $H_2$ | $H_2$ |
| Time on stream, h | 17 | 41 | 89 | 17 | 41 | 40 | 43 | 91 | 145 |
| Temp. °C | 370 | 380 | 380 | 380 | 380 | 380 | 380 | 340 | 320 |
| $C_{28}$ conversion = $100 \times (C_{28}$ in feed $- C_{28}$ in product$)/C_{28}$ in feed | | | | | | | | | |
| $C_{28}$ conversion, % | 78.1 | 75.0 | 68.1 | 3.1 | 5.5 | 6.1 | 99.1 | 29.9 | 14.4 |
| Product distribution, wt% | | | | | | | | | |
| $C_1$ | 0.1 | 0.2 | 0.3 | 0.0 | 0.0 | 0.0 | 2.7 | 0.2 | 0.0 |
| $C_2$–$C_4$ | 0.7 | 2.1 | 2.7 | 0.1 | 0.2 | 0.6 | 27.4 | 4.3 | 0.9 |
| $C_5$–$C_{11}$ | 5.7 | 9.8 | 10.0 | 1.2 | 1.3 | 0.6 | 57.2 | 14.1 | 2.4 |
| $C_{12}$–$C_{19}$ | 28.8 | 37.6 | 28.6 | 1.5 | 1.4 | 1.1 | 10.3 | 8.3 | 2.4 |
| $C_{20+}$ | 64.7 | 50.2 | 58.3 | 97.2 | 97.1 | 97.8 | 2.4 | 73.0 | 94.3 |
| Selectivity, % | | | | | | | | | |
| $C_1$–$C_4$ | 1.4 | 3.1 | 4.6 | 1.7 | 3.9 | 15.2 | 30.8 | 14.1 | 11.2 |
| $C_5$–$C_{11}$ | 10.5 | 13.2 | 15.0 | 24.9 | 28.6 | 16.1 | 58.6 | 44.3 | 31.1 |
| $C_{12}$–$C_{19}$ | 52.5 | 50.4 | 42.9 | 32.2 | 31.7 | 30.0 | 10.5 | 26.1 | 31.5 |
| $C_{20}$–$C_{27}$ | 35.6 | 33.3 | 37.5 | 41.2 | 35.8 | 38.8 | 0.1 | 15.5 | 26.2 |

[a] Other process conditions: 3.2 MPa, $H_2/C_{28} = 20$, LHSV = 2 h$^{-1}$.

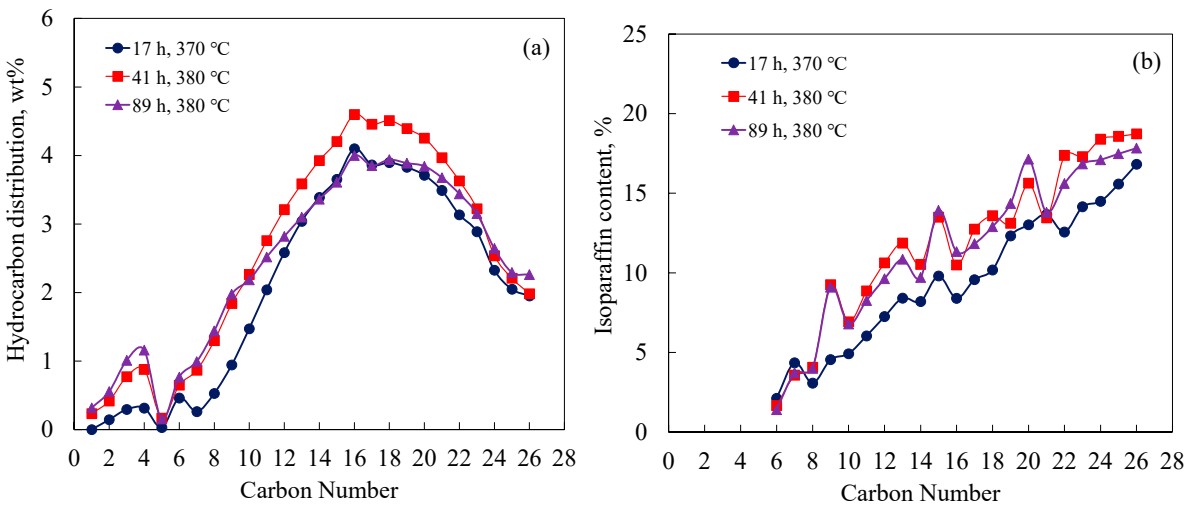

**Figure 6.** (**a**) The product distribution, (**b**) the isoparaffin content (iso/(iso + normal)) for hydrocracking $C_{28}$ over the NiMo/Al activated under $N_2$. (3.2 MPa, $H_2/C_{28} = 20$, LHSV = 2 h$^{-1}$).

According to Table 2, the $N_2$-activated nonsulfided NiMo/Al catalyst displayed 68–78% $C_{28}$ conversion and 29–38% $C_{12}$–$C_{19}$ was in the product, which corresponds to 52–53% selectivity to the middle diesel range hydrocarbons at 370–380 °C in the first 40 h. In contrast, the $H_2$-activated non-sulfided NiMo/Al and the sulfided NiMo/Al catalyst gave poor activity ($C_{28}$ conversion was only ca. 4–6%), and less than 2% $C_{12}$–$C_{19}$ hydrocarbons were identified; however, the $H_2$-activated non-sulfided NiMo/Si-Al was highly active (99% $C_{28}$ conversion) and 57% gasoline range hydrocarbons at the same temperature were obtained. These results clearly indicate that (1) $N_2$-activation is effective to form active nonsulfided NiMo/Al catalyst, and the main products formed are the middle distillates (>50%); (2) the SiAl support (Si/Al = 6.6) is highly active for the hydrocracking reaction, and dominant products formed are gasoline range hydrocarbons (>50%); and (3) the NiMo/Al catalyst pretreated by $H_2$ or $H_2S$ was not an active hydrocracking catalyst.

Furthermore, the results appear to suggest that the catalyst pore size played an important role in controlling product distribution and selectivity, in addition to catalyst acidity. Larger pore sized supports such as NiMo/Al (7.6 nm) tended to yield higher fractions of diesel range hydrocarbons, while the catalyst with relatively small pore size such as the NiMo/SiAl (6.7 nm) tended to produce mainly gasoline range hydrocarbons.

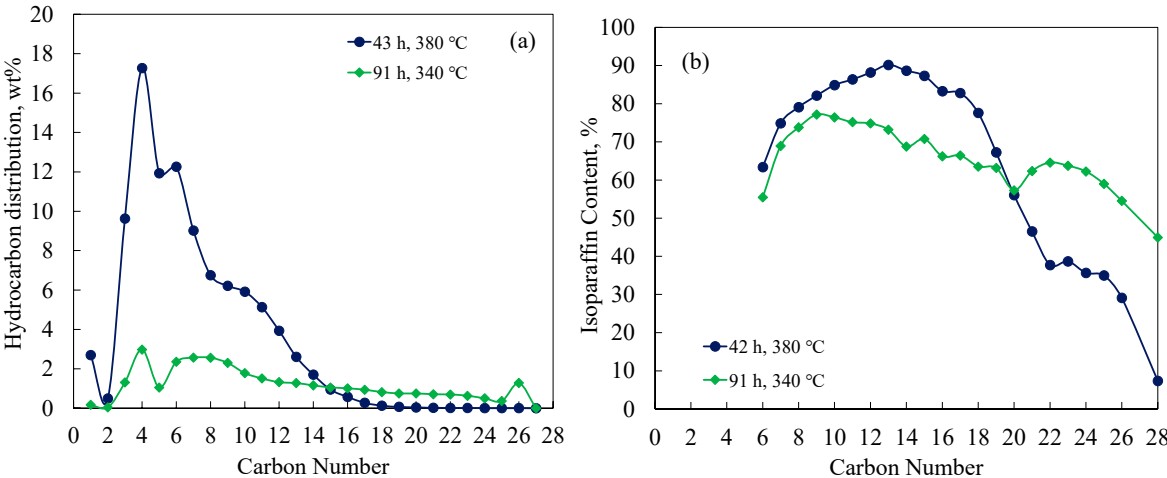

**Figure 7.** (**a**) The product distribution, (**b**) the isoparaffin content (iso/(iso + normal)) for hydrocracking $C_{28}$ over NiMo/Si-Al activated (3.2 MPa, $H_2/C_{28}$ = 20, LHSV = 2 $h^{-1}$).

The hydrocarbon distributions over the NiMo/Al and NiMo/SiAl catalysts are shown in Figures 6a and 7a. Interestingly, the hydrocarbon distributions over the NiMo/Al catalysts (Figure 6a) present nearly a pattern of symmetry, with double peaks at $C_4$ and $C_{16}$ being observed under different conditions. These results imply that C–C bond breakage in the middle of the chain was faster and multiple hydrocracking reactions (e.g., primary and secondary hydrocracking) took place on the catalyst surface ($C_{28} \rightarrow C_{14} \rightarrow C_7 \rightarrow C_4$). The hydrocarbon distributions at 340 °C and 380 °C on the NiMo/Si-Al catalyst (Figure 7a) show a similar pattern but the peak at $C_{16}$ shifted to $C_7$ due to the higher hydrocracking activity of the Si-Al catalyst.

Table 2 shows 0.1–0.3% $CH_4$ and 1–2.7% $C_2$–$C_4$ light hydrocarbons formed on the nonsulfided Ni-Mo/Al ($N_2$-activated), while these numbers increase to 0.8–2.7% and 4.3–27% over NiMo/Si-Al, which represents 1.4–4.6% and 11–31% selectivity to $C_1$–$C_4$ for the two catalysts. The formation of $CH_4$ could indicates that the free radical mechanism played a role; however, the coexistence of the carbonium and free radical mechanism (especially on the NiMo/Si-Al catalysts) could occur. A much higher portion of light hydrocarbons (~27%), significantly greater than the % of $CH_4$ (0.2–2.7%), is noticed.

### 3.3. Hydrocracking FT Wax over Pt/Al and Pt/Si-Al Catalysts

The hydrocracking performance of the Pt/Al and Pt/Si-Al catalysts was examined using FT cobalt wax under similar conditions as that of the NiMo catalysts. A typical hydrocarbon distribution of the FT wax produced using a Pt-Co/$Al_2O_3$ catalyst is shown in Figure 8, which contains about 20% $C_{10}$–$C_{20}$ and 80% $C_{20+}$. The hydrocracking activity of the Pt catalysts was evaluated based on $C_{20+}$ conversion, which is defined in Section 2.6. The hydrocracking results (including activity, hydrocarbon distribution and hydrocarbon selectivity) are summarized in Table 3 and in Figures 9 and 10.

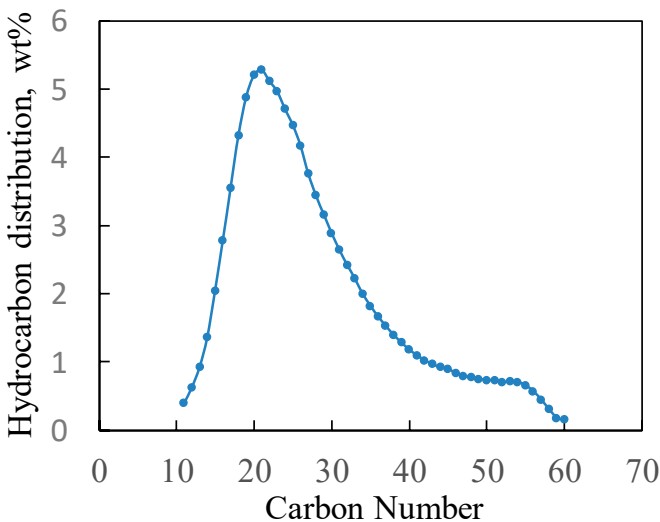

**Figure 8.** Hydrocarbon distribution of Fischer–Tropsch wax produced from a 0.5%Pt-25%Co/Al$_2$O$_3$ catalyst (230 °C, 2.0 MPa, H$_2$/CO = 2.0 and 50% CO conversion).

**Table 3.** Summary of hydrocracking FT cobalt wax on the 0.5%Pt/Al and 0.5%Pt/Si-Al catalysts [a].

| Catalyst | Pt/Al | | | | Pt/Si-Al | | |
|---|---|---|---|---|---|---|---|
| Activation | H$_2$ | | | | | | |
| Time on stream, h | 72 | 90 | 115 | 133 | 94 | 118 | 140 |
| Temp. °C | 320 | 340 | 360 | 380 | 280 | 290 | 300 |
| C$_{20+}$ conversion = 100 × (C$_{20+}$ in feed − C$_{20+}$ in product)/C$_{20+}$ in feed | | | | | | | |
| C$_{20+}$ conversion, % | 10.97 | 25.45 | 42.89 | 32.73 | 39.43 | 82.74 | 99.85 |
| Product distribution, wt% | | | | | | | |
| C$_1$ | 0.16 | 0.21 | 0.34 | 0.48 | 0.02 | 0.03 | 0.03 |
| C$_2$–C$_4$ | 0.10 | 0.47 | 1.29 | 1.76 | 1.45 | 4.23 | 4.71 |
| C$_5$–C$_{11}$ | 0.50 | 1.56 | 3.59 | 4.94 | 2.95 | 32.64 | 63.76 |
| C$_{12}$–C$_{19}$ | 28.24 | 36.61 | 46.43 | 36.42 | 47.61 | 49.73 | 31.38 |
| C$_{20+}$ | 71.00 | 61.15 | 48.34 | 56.39 | 47.96 | 13.36 | 0.12 |
| Selectivity, % | | | | | | | |
| C$_1$–C$_4$ | 1.89 | 1.17 | 1.14 | 2.16 | 0.08 | 0.05 | 0.03 |
| C$_5$–C$_{11}$ | 1.18 | 2.65 | 4.28 | 7.95 | 4.66 | 6.38 | 5.92 |
| C$_{12}$–C$_{19}$ | 4.54 | 8.10 | 11.59 | 21.78 | 8.22 | 48.69 | 79.94 |
| C$_{20}$–C$_{27}$ | 92.40 | 88.07 | 82.98 | 68.11 | 87.05 | 44.87 | 14.11 |

[a] Other process conditions: 3.2 MPa, H$_2$/wax = 20, LHSV = 2 h$^{-1}$.

It is observed from Table 3 that the Pt/Al catalyst displayed 10–43% C$_{20+}$ conversion at 320–380 °C, and the primary products are middle diesel fractions (30–46%), which account for 83–92% of the converted products, with the remaining converted products being less than 0.5% methane and less than 1.8% light hydrocarbons. The C$_{20+}$ conversion at 380 °C was 33% at 134 h, lower than that at 360 °C (43%), which resulted in higher deactivation of the catalyst at higher temperatures. Compared to the Pt/Al catalyst, the Pt/Si-Al catalyst was much more active. It exhibited 40–99% C$_{20+}$ conversion at 280–300 °C. The activity trend on the Pt catalysts is in line with that of the Al and Si-Al supported NiMo catalysts presented above. The most distinctive finding regarding the hydrocracking performance of the Pt/Si-Al catalyst relative to the Pt/Al catalyst is that the product of the catalyst is almost free of CH$_4$, but produced 1.5–4.7% C$_2$–C$_4$ light hydrocarbons. Moreover, the

dominant product range at 290 °C or lower temperature is the middle distillate range of $C_{12}$–$C_{19}$ (48–50%); however, the selectivity shifts toward the gasoline range hydrocarbon range $C_5$–$C_{11}$ (64%) at 300 °C. The difference in the product distribution suggests dissimilar hydrocracking pathways on the Pt/Al and Pt/Si-Al catalysts, which will be discussed in more detail in Section 3.5.

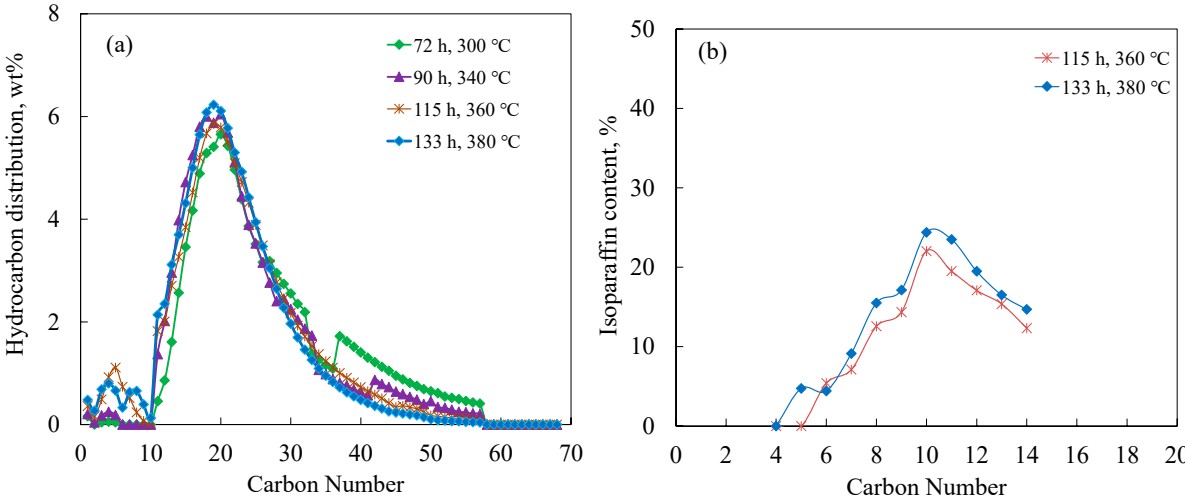

**Figure 9.** (**a**) The product distribution, (**b**) the isoparaffin content (iso/(iso + normal)) for hydrocracking FT wax over Pt/Al activated (3.2 MPa, $H_2$/wax = 20, LHSV = 2 $h^{-1}$).

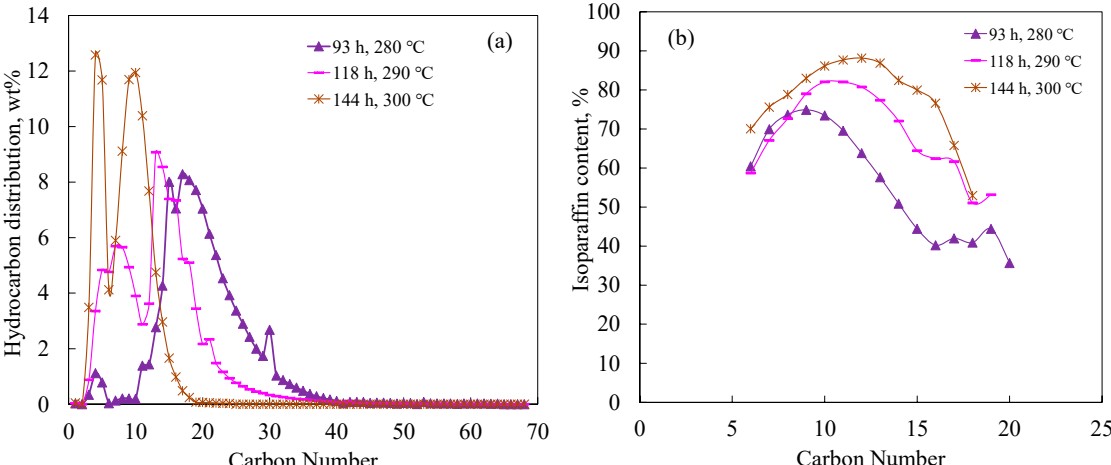

**Figure 10.** (**a**) The product distribution, (**b**) the isoparaffin content (iso/(iso + normal)) for hydrocracking FT wax over Pt/Si-Al activated (3.2 MPa, $H_2$/wax = 20, LHSV = 2 $h^{-1}$).

The product distribution below 43% on both Pt/Al (320–380 °C) and Pt/Si-Al (<280 °C) catalysts presents double peaks at ca. $C_{20}$ and $C_5$ (Figures 9a and 10a). In the case of Pt/Si-Al catalyst, as the reaction temperature was increased to 290 and 300 °C and conversion increased to 83 and 99%, the first peak gradually shifted to lower carbon numbers (i.e., $C_{15}$ and $C_{10}$) due to the high hydrocracking conversion, while the peak at lower carbon range ($C_5$–$C_7$) remains unchanged except for the enhanced intensity (Figure 10a). The results further suggest that secondary hydrocracking occurred on the Pt catalysts.

The $N_2$-activated NiMo/Al catalyst exhibited higher activity than the $H_2$ or $H_2S$/$H_2$-activated NiMo/Al catalyst. This can be associated with higher population of Brønsted acid sites (BAS) generated by the $N_2$ activation. The FTIR pyridine result indicates dominant LAS peaks on all $Al_2O_3$ samples, and the $H_2$-activated Al catalyst showed even higher amounts (higher intensity at 1450 and 1620 $cm^{-1}$) of LAS. Therefore, it can be speculated

that the LAS on the catalyst are not responsible for the high activity of the Al catalyst, but rather the BAS ($1520$ $cm^{-1}$) are the main active sites. The partially oxidized Mo ($MoO_x$) present under the $N_2$-activation might be a type of BAS, thus enhancing the adsorption of the intermediates such as carbonium ions or radical ions on metal sites, and leading to enhanced catalyst performance.

Compared with the activities of the NiMo and Pt catalysts shown in Tables 2 and 3, it is clearly observed that $C_{20+}$ wax conversion on Pt catalyst is higher than $C_{28}$ conversion on the NiMo catalyst regardless of support type. Two major reasons could be responsible for the observed activity discrepancy between the two catalysts. One is that the intrinsic activity of the noble metal Pt is higher than the NiMo catalyst [41]. Another reason could be the dependence of hydrocarbon reactivity on chain length, with it increasing significantly with chain length up to $C_{33}$ [23,42–44]. The wax feed produced from cobalt catalyzed FTS has an average carbon number of $C_{30}$–$C_{31}$; as such, higher conversion is expected relative to the case of pure $C_{28}$ hydrocarbon feed.

### 3.4. Isoparaffin Content versus Carbon Number

To investigate the effect of chain length on isomerization, the isoparaffinic content versus carbon number for the NiMo/Al, NiMo/Si-Al, Pt/Al and Pt/Si-Al catalysts are plotted and shown in Figures 6b, 7b, 9b and 10b. The isomers in the hydrocracking products obtained for NiMo/Al under $H_2$ and $H_2S$ treatment were not analyzed due to the very low conversion obtained. The isomer contents on the Si-Al catalysts are 30–90% (Figures 7 and 10), significantly higher than those (<30%) of the Al catalyst (Figures 6 and 9), which could be due to an adequate balance between the metal and acidity of the Si-Al catalysts [16,17]. Regardless of temperature range (280–380 °C) and hydrocarbons feed (i.e., $C_{28}$ or FT wax) used, the isomer content increases with carbon number, reaching maximum values at $C_{13}$–$C_{14}$ for the NiMo/Si-Al and Pt catalysts (Figures 7b, 9b and 10b). However, the peak values in the product spectrum at lower temperatures on the NiMo/Si-Al (340 °C) or Pt/Si-Al catalyst (280 °C) shifted to a lower carbon number (i.e., $C_{10}$). Afterwards, the isoparaffin content decreased significantly with increasing carbon number. These results are consistent with a study on the hydrocracking on an Fe-based FT wax over a $MoO_3$-Pt/Si-Al catalyst performed by Leckel et al. [23], who reported the peak ratio at $C_{12}$–$C_{13}$ at 370 °C and 5–7 MPa. Interestingly, the change in the isomer content with carbon number for the $N_2$-activated NiMo/$Al_2O_3$ catalyst increases remarkably with increasing carbon number over the entire carbon range, $C_6$–$C_{26}$. This is in line with the study on hydrocracking $C_{16}$ on Pt/HY and Pt-Cs/PV catalysts by Rausch et al. [45], which reported that the iso/normal paraffin ratio increases with chain length in the $C_4$–$C_{12}$ range, and partly consistent with the study of Gamba et al. [10], which reported a similar trend after $C_{10}$ at 343–375 °C on a Pt/Si-Al catalyst (i.e., below $C_{10}$, the isomer fraction decreases with increasing chain length).

Hydrocracking and hydroisomerization are competitive reactions, and as such, they could be kinetically controlled. Thus, the concentration of hydrocracking reaction intermediates and conversion level can significantly impact the isomer/normal product values. According to the hydrocracking product distribution using pure $C_{28}$ as shown in Figure 6a, β-scission of the carbon-carbon bond more likely proceeds faster in the middle of the chain, leading to a near symmetry in the hydrocarbon distribution. This conclusion is probably supported by the study of Gamba et al. [10], which reported that the possibility for the successive isomerization and C–C cracking follows the order: direct alkane hydrocracking < monobranched isomers < di-branched < tri-branch < tetra branched isomers and so on. Since the FT wax distribution used in this study was produced using a cobalt catalysts and centers at $C_{21}$ (Figure 8), the concentration of $C_{10}$–$C_{14}$ isomer intermediates during the hydrocracking of FT cobalt wax or $C_{28}$ could be higher based on the 'symmetrical cracking theory'; therefore, the peak isomer/paraffin ratios occurred at $C_{10}$–$C_{14}$. A decrease in the iso/paraffin ratio with carbon number after the peak value could be due to a spatial effect for the isomerization of large molecules. However, the

isoparaffin content kept increasing with increasing carbon number on the $N_2$-activated NiMo catalyst, and this could be due to (1) an enhanced BAS density on the catalyst; (2) the adsorption energy is increased with increasing chain length [3,46]; (3) heavier hydrocarbons were present mainly in the liquid phase under the reaction conditions, thus resulting in longer chain-length dependent residence times in the catalyst bed, and/or (4) large pore $Al_2O_3$ supports benefit isomerization. These factors could enhance the isomerization of higher molecular weight hydrocarbons. As a result, an increasing trend of the isomerization content with carbon number was observed for this catalyst.

### 3.5. Mechanistic Considerations

The product distribution of the Si-Al supported NiMo and Pt catalysts might indicate the operation of different reaction mechanisms. The most accepted mechanisms are the carbonium and free radical mechanisms. According to the hydrocracking results shown in Table 3, almost no methane was produced at 280–300 °C for the Pt/Si-Al catalyst, which is a clear indication of the cycle following with a carbonium ion mechanism. As discussed in the previous session, the BAS detected on the Pt/Si-Al catalyst by FTIR played a key role in the hydrocracking performance, thus direct splitting of the carbon-carbon bonds of $C_{28}$ or wax alkanes to form shorter hydrocarbons is excluded due to the reaction requiring LAS [47]. As such, the hydrocracking via carbonium intermediates on the Pt/Si-Al catalyst is proposed. As shown in Scheme 1, alkane first dehydrogenates over Pt and generates alkene, which quickly converts to either a carbonium ion via proton addition at BAS (followed by β-scission to form shorter chain hydrocarbons and new carbonium ions), or undergoes another series of processes such as the iso-carbonium ion path. The proposed mechanism would yield little $CH_4$ due to β-scission at the α carbon position not being energetically favorable [3]. In the case of the NiMo/Al and NiMo/Si-Al catalysts, high amounts of $CH_4$ were produced at 320–380 °C (1–5% $CH_4$ selectivity). Thus, the free radical mechanism is proposed to account for the majority of hydrocarbon formation, while a small portion could conceivably be produced from thermal cracking. As indicated in Scheme 2, the initiation step involves the C–C bonding electron pair being split by homolytic bond fission between the two formed alkyl radicals followed by termination steps to form various normal products and isomers. However, the radical recombination, addition, radical reactions and/or formation of carbonium ions could also take place, due to the high selectivity of $C_2$–$C_4$; thus, the carbonium ion mechanism might also play in the hydrocracking reactions on NiMo catalysts, especially on the Si-Al supported catalyst (Scheme 2).

**Scheme 1.** Proposed carbonium ion mechanism for hydrocracking FT cobalt wax on Pt/Si-Al.

$$n\text{-}C_{28}\,(C_n) \;\longrightarrow\; C_{n\text{-}y}H_{(2(n\text{-}y)+1)}\bullet + C_yH_{(2y+1)}\bullet \quad y = 1, 2, 3\ldots n\text{-}1$$

$$C_{n\text{-}y}H_{(2(n\text{-}y)+1)}\bullet \;\xrightarrow{\;+H_2\;}\; C_{n\text{-}y}H_{(2(n\text{-}y)+2)}$$

$$C_yH_{(2y+1)}\bullet \;\xrightarrow{\;+H_2\;}\; C_yH_{(2y+2)}$$

$$C_yH_{(2y+1)}\bullet + C_{(n\text{-}y)}H_{(2(n\text{-}y)+2)} \;\longrightarrow\; C_yH_{2y+2} + C_{(n\text{-}y)}H_{(2(n\text{-}y)+1)}\bullet \quad y = 1, 2, 3\ldots n\text{-}1$$

$$C_nH_{(2n+1)}\bullet \;\xrightarrow{\text{Alky shift}}\; C\text{-}C\text{-}C\ldots C\text{-}C_{n\text{-}1}\bullet \;\xrightarrow{\;+H_2\;}\; C\text{-}C\text{-}C\ldots C\text{-}C_{n\text{-}1} \quad n = 1, 2, 3\ldots$$

$$\downarrow\qquad\qquad\qquad\downarrow\qquad\qquad\qquad\downarrow$$

$$CH_3\qquad\qquad CH_3$$

$$H\bullet + C_nH_{2n}\;\; \xrightarrow{\text{Carbonium, alky shift \& }+H_2}$$

**Scheme 2.** Proposed hybrid free radical and carbonium hydrocracking mechanisms for $C_{28}$ on NiMo/Al or NiMo/Si-Al.

## 4. Conclusions

The hydrocracking of $C_{28}$ model compound over a commercial NiMo/Al$_2$O$_3$ catalyst was studied in order to shed light on the influence of activation environment, i.e., $N_2$, $H_2$, and $H_2S/H_2$. A NiMo/Si-Al catalyst and a Pt based Al and Si-Al catalysts were also evaluated in order to comparatively study activity, product production, isomer content as a function of carbon number, and hydrocracking pathways on the catalysts. Important results and insights on the hydrocracking were obtained.

The SEM and XRD experimental results demonstrated highly dispersed metals (Ni, Mo, or Pt) on the rough surface of the Al$_2$O$_3$ and amorphous Si-Al supports. The FTIR and XRD studies evidenced that $N_2$-activation generated partially oxidized Mo (MoO$_x$, x = 2–3) and higher amounts of BAS on the Al$_2$O$_3$ support. As a consequence, higher hydrocracking activity was obtained relative to the catalysts treated with $H_2$ and $H_2S/H_2$. The Si-Al supported NiMo and Pt catalysts displayed very high activity and high isomer content, which is ascribed to the relatively higher surface densities of BAS on the acidic support.

Hydrocracking of $C_{28}$ on the nonsulfided NiMo/Al catalyst is proposed to proceed in a symmetrical manner and β-scission likely took place faster in the middle of the hydrocarbon chain. Secondary hydrocracking was evidenced by the double-peak product distribution for the hydrocracking of $C_{28}$ (at $C_4$ and $C_{16}$) and cobalt FT wax (at $C_5$ and $C_{10}$). Increasing temperature shifted the product distribution to lower hydrocarbons.

The nonsulfided NiMo/Al catalyst ($N_2$ activated) and Pt/Al catalysts produced mainly diesel range hydrocarbons (sel. > 50%), while the NiMo/SiAl and Pt/Si-Al catalysts produced predominantly gasoline range (sel. > 50%) hydrocarbons. The results appear to reflect support effects (e.g., pore size and acidity). However, lower temperatures (<300 °C) shifted the product distribution of the Pt/Si-Al catalyst to mainly diesel hydrocarbons. Methane (0.1–2.7%) and light hydrocarbons $C_2$–$C_4$ (0.7–27%) were produced on all NiMo and Pt/Al catalysts; however, the Pt/Si-Al produced virtually negligible CH$_4$ under the tested conditions. A carbonium ion mechanism and a free radical plus carbonium hybrid mechanism that are likely associated with support (Al, Si-Al), metal (Ni, Mo, Pt) and process conditions, were proposed to account for differences in the product distribution. For example, the hydrocracking on Pt/Si-Al below 300 °C appears to follow a carbonium ion mechanism, while all other catalysts may follow the hybrid mechanism.

The hydrocracking and hydroisomerization are competitive reactions that could be kinetically and thermodynamically controlled. This resulted in a dependence of isomer content (iso-alkane/total alkane) on hydrocarbon chain length. It was postulated that factors such as the spatial effect during isomerization, varied residence time of hydrocarbons in the liquid phase with chain length due to vapor-liquid equilibrium, reactivity ($k_{iso}$) differences based on chain length, and support acidity could influence isomer formation.

Using the Si-Al support with small pores, the peak isomer contents occurred at $C_{10}$–$C_{13}$ during hydrocracking $C_{28}$ or FT wax (produced using a Pt-Co catalyst), but an exception was observed when hydrocracking $C_{28}$ on the NiMo/Al with relatively large pore, in which the spatial effect might be weakened; as such, isomerization was enhanced as a function of carbon number.

**Author Contributions:** Conceptualization, methodology, investigation, data analysis, writing, validation—W.M. and J.K., Investigation, review, editing and validation, G.J. and S.D.H., Supervision, B.H.D. All authors have read and agreed to the published version of the manuscript.

**Funding:** This research received no external funding.

**Acknowledgments:** This paper is dedicated to the memory of the late Burtron H. Davis, whose ideas, dedication, and supervision were an inspiration to us all. The authors would like to acknowledge the support of the Commonwealth of Kentucky. Liquid analysis by Robert Keogh is greatly acknowledged.

**Conflicts of Interest:** The authors declare no conflict of interest.

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
