# Peer review of "Hydrocracking of Octacosane and Cobalt Fischer–Tropsch Wax over Nonsulfided NiMo and Pt-Based Catalysts"

_reactions, doi:10.3390/reactions2040024_

Round 1
Reviewer 1 Report
The manuscript is well-written and offers some intertesting results in the Hydrocracking of Octacosane and Cobalt Fischer-Tropsch Wax. I recommened the manuscript to Reactions after minor revisions.
(1) Title
Change "Nonsulfided Nimo" to "Nonsulfided NiMo".
(2) Abstract (Line 18)
Change "...to higher amounts of BAS on the catalyst." to "...to higher amounts of Brønsted acid sites (BAS) on the catalyst."
(3) Keywords
The number of Keywords (15) in manuscript is too much. Please cut some Keywords to make the number below 10 according to Instructions for Authors.
(4) Introduction
Fig. 1 should be deleted. In gerenal, the Introduction Part just explain results in literatures but do not use the figures in references. In paticular, the data in Fig 1 (Ref. 10, 23) are not the authors' works. The use of Fig 1 in the manuscript has a problem of copy right.
(5) In Line 43
Change "AIMCM-41 zeolite" to "AlMCM-41 zeolite".
(6) In Line 203 and 212:
Change "C5-C11 = 100× C4-C11 in product/..." to "C5-C11 = 100× C5-C11 in product/...".
(7) Shift Fig. 10 to about Line 367. It is better to introduce the composition of the FT wax feedstock before the results of hydrocracking of FT wax.
The composition of Co-besed catalyst (Co loading, support) and reaction conditions for the FT reaction (temperature, pressure) should be described. The temperature in the distillation of FT product to obtain FT wax also should be described.
(8) The form of References is not suable for Reactions. Please change them according to Instructions for Authors.
(9) In Line 585, Ref. 17:
Change "in Recent Progress in Mesostructured Materials Proceedings of 585
the 5th International Mesostructured Materials Symposium (IMMS2006), Shanghai, P.R. China, August 5-7, 2006 Editors D.Y. 586
Zhao, S.L Qiu, Y. Tang, C.Z. Yu, v 165, Pages 1-922 (2007). "
to "in Recent Progress in Mesostructured Materials Proceedings of 585
the 5th International Mesostructured Materials Symposium (IMMS2006), Book Series: Studies in Surface Science and Catalysis Volume: 165 Pages: 781-785 Published: 2007."
Author Response
The manuscript is well-written and offers some intertesting results in the Hydrocracking of Octacosane and Cobalt Fischer-Tropsch Wax. I recommend the manuscript to Reactions after minor revisions.
Response: We appreciate reviewer’s favorable comment.
- Title Change "Nonsulfided Nimo" to "Nonsulfided NiMo".
Response: In our original document, it is NiMo, means “Mo” begins with capital “M”, not “m”, the reviewer is correct that NiMo should be retained in the final version.
(2) Abstract (Line 18)
Change "...to higher amounts of BAS on the catalyst." to "...to higher amounts of Brønsted acid sites (BAS) on the catalyst."
Response: Done. Please see the change on page 1. We also changed the words "to higher amounts of" to "to a higher number of" for clarity.
(3) Keywords
The number of Keywords (15) in manuscript is too much. Please cut some Keywords to make the number below 10 according to Instructions for Authors.
Response: We thank the Reviewer for the comment. We have reduced the number of keywords to 9. Please see the change on Page 1.
(4) Introduction
Fig. 1 should be deleted. In general, the Introduction Part just explain results in literatures but do not use the figures in references. In particular, the data in Fig 1 (Ref. 10, 23) are not the authors' works. The use of Fig 1 in the manuscript has a problem of copy right.
Response: First, we have obtained the copyright permissions for the two plots from the publishers of the two original papers; they are attached. The purpose of using the two plots here is to describe the background of this study so that the readers can have a much clearer picture of the different results obtained in the literature. This certainly is helpful to bring about one research objective we focused on in this paper. We believe that there are no copyright issues for using Figure 1 to help provide background in the context of this study.
(5) In Line 43
Change "AIMCM-41 zeolite" to "AlMCM-41 zeolite".
Response: done, changed "AIMCM-41 zeolite" to "AI-MCM-41 zeolite", please see page 1.
(6) In Line 203 and 212:
Change "C5-C11 = 100 × C4-C11 in product/..." to "C5-C11 = 100 × C5-C11 in product/...".
Response: We appreciate the correction by the reviewer. Done. Please see the changes on page 6.
(7) Shift Fig. 10 to about Line 367. It is better to introduce the composition of the FT wax feedstock before the results of hydrocracking of FT wax.
The composition of Co-besed catalyst (Co loading, support) and reaction conditions for the FT reaction (temperature, pressure) should be described. The temperature in the distillation of FT product to obtain FT wax also should be described.
Response: We have changed Figure 10 to Figure 8, and move the plot to about line 385 as suggested. Consequently, the original Figures 8 and 9 were changed to Figures 9 and 10, respectively. The cobalt catalyst composition, reaction conditions used for obtaining the cobalt wax are provided. Please see the changes between page 12 and page 14.
(8) The form of References is not suable for Reactions. Please change them according to Instructions for Authors.
Response: All references have been formatted to match the formatting requirements of the journal.
(9) In Line 585, Ref. 17:
Change "in Recent Progress in Mesostructured Materials Proceedings of 585
the 5th International Mesostructured Materials Symposium (IMMS2006), Shanghai, P.R. China, August 5-7, 2006 Editors D.Y. 586
Zhao, S.L Qiu, Y. Tang, C.Z. Yu, v 165, Pages 1-922 (2007). "
to "in Recent Progress in Mesostructured Materials Proceedings of 585
the 5th International Mesostructured Materials Symposium (IMMS2006), Book Series: Studies in Surface Science and Catalysis Volume: 165 Pages: 781-785 Published: 2007."
Response: Changed. Please see the changes on page 19.
Reviewer 2 Report
The study of catalysts for hydrocracking and isomerization of long-chain paraffinic hydrocarbons is a current topic, as is the effect of the atmosphere used during activation. The paper is well organized and the results are interesting. However there are some aspects that need to be better clarified.
First of all it is necessary to better explain what was the design of experiments, and why in the case of NiMo/Al were chosen the activation with N2 and H2/H2S and in the case of NiMO/AlSi only the activation with H2 was adopted.
In table 2 it appears that the NiMo/AlSi is very sensitive to the TOS. The C28 conversion goes from 6.1 to 99.1 % in just three hours (apparently at the same operating conditions), is there any explanation for this sharp increase in hydrocracking activity?
Because the temperature of 380°C is quite high, a blank test to determine the effect of this temperature on C28 hydrocarbons (or on FT mixture) can be useful for determining the amount of thermal cracking.
For the test made with wax from FT the hydrocarbon distribution (and eventually the amount of isomers) of the charge is not reported
Some minor commets:
In the title there is a typo, Nimo instead NiMo)
Please define in the test what is TOS.
Lines 391-399 do not seem well placed in the text. Probably they should be moved to the previous section
Author Response
The study of catalysts for hydrocracking and isomerization of long-chain paraffinic hydrocarbons is a current topic, as is the effect of the atmosphere used during activation. The paper is well organized, and the results are interesting. However, there are some aspects that need to be better clarified.
Response: The authors appreciate the reviewer's favorable feedback of our work.
First of all it is necessary to better explain what was the design of experiments, and why in the case of NiMo/Al were chosen the activation with N2 and H2/H2S and in the case of NiMO/AlSi only the activation with H2 was adopted.
Response: The reason why we focused on improving performance of the NiMo/Al catalyst by changing the activation environment is because our recent results showed poor activity of the catalyst under normal H2 activation, in contrast, the NiMo/Si-Al catalyst system was very active for the hydrocracking of hydrocarbons after the normal H2 activation due to the acidic nature of the catalyst. Therefore, it does not significantly change the performance of the Si-Al catalyst by using other activation methods.
In table 2 it appears that the NiMo/AlSi is very sensitive to the TOS. The C28 conversion goes from 6.1 to 99.1 % in just three hours (apparently at the same operating conditions), is there any explanation for this sharp increase in hydrocracking activity?
Response: The activity of the NiMo/AlSi changed significantly from 99 to 14% between 44 to 145 h, and this was due to the significantly varying reaction temperature from 380 oC to 320 oC. Other reaction conditions were maintained the same (3.2 MPa, H2/C28 = 20, LHSV = 2 h-1). Please see the values listed in Table 2 on Page 11.
Because the temperature of 380°C is quite high, a blank test to determine the effect of this temperature on C28 hydrocarbons (or on FT mixture) can be useful for determining the amount of thermal cracking.
Response: We thank the Reviewer for the comment. The NiMo/Al catalyst was not active and requires high temperatures (e.g., 340-400 oC) for the hydrocracking reaction to be suitably active. We did not conduct thermal cracking of C28 and wax at 380 oC, but assuming that a portion of the CH4 produced is from thermal cracking, the thermal cracking activity of the NiMo/Al catalyst at 380 oC would be very low due to the small amounts of CH4 produced (Table 2, 0.1-0.3%), while the thermal cracking activity of the NiMo/Si-Al catalyst at 380 oC would be relatively higher (Table 2, CH4% = 2.7%). The possibility of thermal cracking is also discussed in Section 3.5 (mechanistic considerations) on Page 16.
For the test made with wax from FT the hydrocarbon distribution (and eventually the amount of isomers) of the charge is not reported
Response: The hydrocarbon distribution of cobalt wax feed is shown in Figure 8, while the hydrocarbon distribution of products over Pt/Al and Pt/Si-Al catalysts are shown in Figures 9 and 10. The cobalt FT wax feed is virtually isomer-free, while the distribution of the isomers on the NiMo/Al catalyst at low temperatures are small and have larger error. As such, they are not shown, while the isomer results at higher temperatures (e.g., 360 and 380 oC) were distinct and are provided in Figure 9.
Some minor commets:
In the title there is a typo, Nimo instead NiMo)
Response: We thank the reviewer for catching this. In our original manuscript submitted, it is correctly shown as NiMo.
Please define in the test what is TOS.
Response: We appreciate the reviewer's suggestion. We have changed TOS to Time on stream
Lines 391-399 do not seem well placed in the text. Probably they should be moved to the previous section
Response: We thank the reviewer for the comments. We have checked the original text (lines 391-399), and they appear to be okay in the revision.
Reviewer 3 Report
The manuscript studies the hydrocracking of Octacosane and Fischer-Tropsch Wax 2 over non-sulfided NiMo- and Pt-based catalysts. The authors have already conducted extensive research on Pt-based catalysis for hydrocracking conversion. However, in this new contribution, they have now studied the effect of the activation environment (N2, H2, and H2S/H2) on the hydrocracking performance of a new catalyst (NiMo/Al), and compared that with the reference Pt/Si-Al and Pt/Al catalysts for hydrocracking wax. The materials were also extensively characterized by BET, XRD, SEM-EDX, and FTIR techniques. Overall, the work is a new contribution to the field and fits well to the Special Issue "Catalytic Conversion of Carbonaceous Materials to Fuels and Chemicals". It can be accepted as it is.
Author Response
The manuscript studies the hydrocracking of Octacosane and Fischer-Tropsch Wax 2 over non-sulfided NiMo- and Pt-based catalysts. The authors have already conducted extensive research on Pt-based catalysis for hydrocracking conversion. However, in this new contribution, they have now studied the effect of the activation environment (N2, H2, and H2S/H2) on the hydrocracking performance of a new catalyst (NiMo/Al), and compared that with the reference Pt/Si-Al and Pt/Al catalysts for hydrocracking wax. The materials were also extensively characterized by BET, XRD, SEM-EDX, and FTIR techniques. Overall, the work is a new contribution to the field and fits well to the Special Issue "Catalytic Conversion of Carbonaceous Materials to Fuels and Chemicals". It can be accepted as it is.
Response: The authors appreciate the reviewer's favorable comments on our manuscript.
Round 2
Reviewer 2 Report
The AA addressed all the question of the referee. Table 2 was reformatted and is now more clear to which catalyst the reported results should be attributed.